# Quality of Life in Teduglutide-Treated Patients with Short Bowel Syndrome Intestinal Failure—A Nested Matched Pair Real-World Study

**DOI:** 10.3390/nu15081949

**Published:** 2023-04-18

**Authors:** Elisabeth Blüthner, Ulrich-Frank Pape, Frank Tacke, Sophie Greif

**Affiliations:** 1Charité—Universitätsmedizin Berlin, Corporate Member of Freie Universität Berlin and Humboldt-Universität zu Berlin, Medical Department, Division of Hepatology and Gastroenterology, Campus Virchow-Klinikum and Campus Charité Mitte, 10117 Berlin, Germany; ul.pape@asklepios.com (U.-F.P.); frank.tacke@charite.de (F.T.); sophie.greif@charite.de (S.G.); 2Berlin Institute of Health (BIH), 10178 Berlin, Germany; 3Department of Internal Medicine and Gastroenterology, Asklepios Klinik St. Georg, Asklepios Tumorzentrum Hamburg ATZHH, 20099 Hamburg, Germany

**Keywords:** quality of life, SF-36, SBS-QoL^TM^, teduglutide, glucagon-like peptide-2, parenteral nutrition, stool frequency, sleep disturbances

## Abstract

Background: Quality of life (QoL) data of chronic intestinal failure (cIF) patients treated with the GLP-2 analogue teduglutide are scarce. This study aims to analyze QoL changes over time in teduglutide-treated patients and compare the results to a matched non-treated cIF control group in a real-world setting. Methods: QoL data (SF-36 and SBS-QoL^TM^) were obtained from adult cIF patients being treated with teduglutide and compared to previously collected QoL data from a PNLiver trial (DRKS00010993), during which patients had been therapy naive. The dataset was then extended by a pairwise matched control group (non-teduglutide-treated PNLiver trial patients) and follow-up data from this group were collected accordingly. Results: Median teduglutide treatment duration and the follow-up period of controls were both 4.3 years. SBS-QoL^TM^ subscales and the SBS-QoL^TM^ sum score showed significant improvements over time in teduglutide-treated patients, as well as for the SF-36 physical and mental component summary scores (all *p* < 0.02), while non-treated patients showed no significant changes in any of the mentioned scores. Significant differences of QoL changes between treated and non-treated patients were seen for both SF-36 summary scores (*p* = 0.031 and 0.012). Conclusions: We herein demonstrate for the first time that QoL significantly improved during teduglutide treatment in SBS-cIF patients in a real-world setting compared to individually matched non-treated SBS-cIF patients, indicating relevant clinical benefits.

## 1. Introduction

Short bowel syndrome (SBS) is a malabsorptive disorder mostly caused by surgical interventions, which may result in chronic intestinal failure (cIF). Parenteral support (PS, i.e., intravenous solution with or without macronutrients) is still the mainstay of treatment for patients with irreversible cIF. New hormonal therapies, such as the glucagon-like peptide-2 (GLP-2) analogue teduglutide, promote mucosal growth and intestinal absorption, thereby leading to a consequent reduction in parenteral support and symptoms, e.g., related to large stoma or fecal losses [1].

Patients with cIF were shown to suffer from psychological problems related to their PS, a disturbed social life and somatic problems [2,3]. Despite considerable advances in surgical technique and introduction of novel targeted therapies, improvement of quality of life is still the major therapy goal in this chronically ill patient population. Taken into consideration that Jeppesen et al. had shown that QoL in SBS patients is comparable with impaired QoL in patients with chronic renal failure treated by dialysis [4] and Blüthner et al. revealed no overall improvement in QoL during the last 20 years [5], QoL still receives proportionally little attention, even though QoL assessment is an essential part of assessing patient-relevant endpoints, particularly in the approval of orphan drugs. Of note, in the clinical phase III STEPS trial, which lead to authority approval, Jeppesen et al. have previously shown a significant improvement of the SBS-QoL total score after 24 weeks of treatment with teduglutide but could not identify a statistically significant difference between treatment and placebo group [6]. Accordingly, Chen and colleagues carried out a post hoc analysis of the aforementioned clinical trial data based on 86 patients, confirmed the results of Jeppesen et al. and pointed out statistical differences for specific SBS-cIF subgroups [7]. Taken together, these clinical trial data showed some benefit of teduglutide treatment on QoL, but real-life data on QoL changes during teduglutide treatment are not available yet. During the release of teduglutide in Germany in 2014, we established the PNLiver trial (DRKS00010993) that recruited chronic intestinal failure patients with parenteral nutrition (PN, i.e., macronutrient containing admixture) from 2014 till 2019 to evaluate the capability of non-invasive liver function tests in cIF patients in a cross-sectional (*n* = 90) and longitudinal (*n* = 20) study [8]. All participants underwent study visits including clinical examination, dynamic liver function assessment, comprehensive blood tests, nutritional status assessment and quality of life assessment with Short Form 36 [SF-36] and SBS-QoL^TM^ [5]. 

The comprehensive dataset of a large monocentric cIF cohort prior to teduglutide treatment from the PNLiver trial enables a follow-up assessment for an observation period far beyond the previously described 24 weeks of patients who were, in the meantime, exposed to teduglutide in a real-life setting. As QoL tends to improve with longer PS duration [9], we extended the results with a pairwise matched control group from non-teduglutide-treated PNLiver trial patients and collected follow-up data from this group accordingly.

## 2. Materials and Methods

### 2.1. Study Design

The study design type is a nested matched pair analysis: Participants of the PNLiver trial (DRKS00010993) underwent study visits including clinical examination, dynamic liver function assessment, comprehensive blood tests, nutritional status assessment and quality of life assessment with Short Form 36 [SF-36] and SBS-QoL^TM^ [5]. This comprehensive dataset was used as baseline data. A follow-up assessment was performed prospectively resulting in a group of patients who were treated with teduglutide and the other group of patients who were not treated with teduglutide and served as controls. If several study visits were available from the PNLiver trial, the follow-up period of controls was chosen to be as close to the treatment duration of the teduglutide-treated patients.

### 2.2. Study Population

Adult patients who took part in the PNLiver trial were enrolled if they were teduglutide-treated or if they were identified as a non-treated best matching partner. The former PNLiver trial included chronic intestinal failure patients receiving parenteral nutrition without underlying liver disease. Patients without quality of life data at baseline, lost to follow-up, teduglutide treatment at baseline/teduglutide discontinuation, or life-changing events during the follow-up period were excluded from this study (see the study flow chart, Figure 1). Remnant bowel anatomy was categorized as group 1 (jejunostomy or ileostomy, 0% colon in continuity), group 2 (jejunocolic anastomosis) and group 3 (jejuno-ileo-colonic-anastomosis) [10]. The onset date of chronic intestinal failure was defined as the last surgical intervention which led to PS dependency. Teduglutide therapy was initiated in PS-dependent stable short bowel syndrome patients who agreed to therapy initiation and who had no contraindications such as active or suspected malignancy, history of malignancy in the gastrointestinal tract within the last five years, suspicious colorectal polyps or recurrent intestinal obstructions in the past.

### 2.3. Matching

An individual matching approach, with matching subject by subject, was performed. As potential confounders, the following variables were selected to differ as minimally as possible between teduglutide-treated patients and controls (ranked order): PS duration before baseline visit, QoL scores at baseline, time to follow-up, PS burden (represented by PN days per week and PS volume per week), bowel anatomy. In case of two eligible matching partners, age, sex, and employment type were also considered.

### 2.4. Ethics and Informed Consent

The study was performed in accordance with the Helsinki Declaration of 1975, as revised in 1983. Ethics committee approval was obtained (EA1/115/20) and written informed consent for study participation was obtained before inclusion in the study. 

### 2.5. Parenteral Support

Parenteral support (PS) information was taken from medical records and is defined as either both fluid and electrolytes alone (intravenous fluid, IVF) or as macronutrients containing admixture (parenteral nutrition, PN). 

### 2.6. Stool Characteristics

Stool frequency was documented per 24 h and stool consistency was classified by patients into five categories: liquid, mushy, soft blobs, firm and constipation, which correlated to Bristol Stool Form Scale categories 7, 6, 5, 4 to 2 and 1, respectively [11]. 

### 2.7. Sleep Disturbances

Sleep disturbances were not documented during the PNLiver trial but for follow-up assessment, patients were asked for their current number of sleep disturbances per night. 

### 2.8. Bioelectrical Impedance Analysis

Measurement of body composition was performed with a bioelectrical impedance analyzer (Nutriguard-M, Data Input, Pöcking, Germany) as described elsewhere [12].

### 2.9. Blood Parameters

Citrulline concentration was determined by high-pressure liquid chromatography in the central clinical chemistry laboratory with normal reference values from 12 to 55 µmol/L. 

### 2.10. Quality of Life Assessment

Two self-completed questionnaires were used for QoL assessment. The SF-36 health status questionnaire is not disease-specific and includes 36 items divided into 8 domains (physical functioning, role-physical, bodily pain, general health, vitality, social functioning, role-emotional and mental health), as well as a domain-based physical and mental health summary scale [13]. The scores range from 0 to 100, with higher scores representing better QoL, and were calculated using HTS 5 (Hogrefe, Göttingen, Germany). 

The SBS-QoL^TM^ questionnaire was developed and validated especially for short bowel syndrome patients with or without PS. It comprises 17 items including two subscales, with lower scores representing better QoL. It is recommended to measure treatment-induced changes in QoL over time in subjects with SBS [14].

### 2.11. Statistical Analysis

Statistical analyses were performed using IBM SPSS Statistics for Windows, Version 28.0 (IBM Corporation, Armonk, NY, USA). Metric values were tested for normal distribution using the Shapiro–Wilk test justifying parametric statistical testing for this analysis. Differences of follow-up vs. baseline values were analyzed using Student’s paired *t*-test or Wilcoxon signed-rank test, differences between groups were analyzed using Student’s *t*-test or Mann–Whitney-U test. The relationship between PS changes and QoL score changes were analyzed by nonparametric Spearman’s rank correlation coefficient. For teduglutide-treated patients, a one-tail *p*-value < 0.05 and for non-treated patients a two tail *p*-value was considered statistically significant. Our primary hypothesis was that teduglutide-treated patients would report an improved QoL compared to non-treated patients. Values are given as means ± SD or as medians and interquartile range (IQR), as indicated.

## 3. Results

### 3.1. Study Cohort

All patients who were enrolled in the PNLiver trial and had started teduglutide treatment after PNLiver trial baseline data collection were eligible for study inclusion. A total of 32 teduglutide-treated patients were identified. Before follow-up for the present study could be performed, six patients died and in five patients, teduglutide was discontinued due to the following reasons: pre-neoplastic adenoma development, insufficient therapeutic effect, on patient’s request due to possible related side effects in three patients. Further, two patients were exposed to teduglutide already at a PNLiver trial baseline visit, one patient developed terminal renal failure, and for one patient, baseline QoL data were missing; therefore, 17 teduglutide-treated patients in total were enrolled. Another 17 non-treated patients were selected by individual matching and allocated to the control group (Figure 1).

Baseline patient characteristics of teduglutide-treated vs. non-teduglutide-treated patients (control group) are shown in Table 1. Baseline data obtained from the PNLiver trial were collected from 2014 to 2019, and follow-up data obtained exclusively for the present study were collected from 2020 to 2022. The administered teduglutide dose was 0.05 mg/kg body weight in all patients (*n* = 17) who were treated with teduglutide. In 10 patients, the teduglutide dose was interrupted temporarily or reduced temporarily to 50% (or both) due to well-known adverse events. Seven patients returned to full dose with considerable time before follow-up, two patients stayed at 50% dose reduction until follow-up (reasons were undesired body weight gain and recurrent hypoglycemia) and in one patient, the dose was further reduced to 50% every other day before follow-up due to undesired weight gain.

### 3.2. Matching Characteristics

Group characteristics after individual matching are shown in Table 2. No statistically significant differences between the teduglutide and control group were found.

### 3.3. Quality of Life

Quality of life was analyzed with SF-36 and SBS-QoL^TM^ questionnaires. SBS-QoL^TM^ subscales 1 + 2 and SBS-QoL^TM^ sum score showed significant changes over time in teduglutide-treated patients (the lower the score, the better the QoL), as well as SF-36 physical and mental component summary scores (the higher the score, the better the QoL), while non-treated patients showed no significant changes in any of the mentioned scores (Figure 2 and Table 3).

QoL change over time in teduglutide-treated vs. non-treated patients (between-group comparison) showed statistical significance in both SF-36 summary scales (*p* = 0.031 and *p* = 0.012, Table 3).

### 3.4. Clinical Parameters

Parenteral support (PS) was analyzed in both groups from baseline to follow-up (Figure 3). Mean PS volume (Δ −7.6 L vs. Δ −4.3 L), PN energy (Δ −5018 kcal vs. Δ −2659 kcal) and PN days per week (Δ −3.3 vs. Δ −1.7) were significantly reduced in both teduglutide-treated and non-treated patients over time. IVF days were significantly reduced in teduglutide-treated patients vs. controls (Δ −1.0 vs. Δ −1.2). In the teduglutide group and in the control group, 4 and 2 out of 17 patients were weaned completely off PS, respectively.

Correlation analyses of the reduction in PS volumes (absolute and relative) and SBS-QoL^TM^ sum score, as well as SF-36 physical and mental component summary scores, were non-significant, respectively.

Stool frequency decreased significantly in teduglutide-treated patients (Δ − 3.8 times per day, *p* = 0.001) and increased significantly in non-treated patients (Δ + 1.6 times per day, *p* = 0.027). Between-group comparison showed that the changes in stool frequencies differ significantly from each other (*p* < 0.001). Stool consistency improved in 5 of 17 (29%) teduglutide-treated patients, while it worsened in 4 from 15 (27%) in non-treated patients (Figure 4).

Citrulline levels increased in teduglutide-treated patients from 23.22 µmol/L to 42.19 µmol/L (*p* = 0.002) and did not show a significant change in non-treated patients (25.82 vs. 26.75 µmol/L) as presented in Figure 5.

Bioelectrical impedance analysis parameters (body cell mass, phase angle α, resting energy expenditure, extracellular mass/body cell mass—Index) did not show any significant changes over time (data not shown).

Sleep disturbances were not collected within the PNLiver trial so data were missing at baseline but collected for follow-up. There was a statistical difference in sleep disturbances between groups: teduglutide-treated patients had less sleep disturbances (median 1.5 (2.8), *n* = 13) compared to non-treated patients (median 2.8 (1.4), *n* = 12) at follow-up (*p* = 0.049).

## 4. Discussion

It is well-known that QoL is impaired in patients with PS-dependent SBS-IF. QoL analyses of clinical trial data in teduglutide-treated SBS-IF patients provided first evidence of a potential QoL-improving effect of teduglutide treatment (6, 7). In the present study, this hypothesis was confirmed, and we herein demonstrate, for the first time in a real-world setting, that QoL scores improved during teduglutide treatment on a statistically significant level compared to non-treated, matched SBS-cIF patients. Due to the reductions in PS requirements, improvements in stool characteristics and lower number of sleep disturbances in teduglutide-treated patients, we provide evidence supporting that these changes are clinically significant as well.

Both applied questionnaires, the SF-36 and the disease-specific SBS-QoL^TM^ scale, showed significant improvements in QoL scores over time in teduglutide-treated patients, while non-treated patients showed no significant changes (Figure 2 and Table 3). Further, SF-36 QoL scores improved significantly from baseline to follow-up in teduglutide-treated vs. non-treated patients in both physical and mental component summary scores (between-group comparisons). In comparison to the previously reported SF-36 results from initial phase III trial, in which the QoL scores from baseline to week 24 indicated no major teduglutide effect on QoL [15], the present results show, for the first time, a positive effect of teduglutide on QoL compared to a non-treated cohort. However, between-group SBS-QoL^TM^ sum score changes did not reach statistical significance (*p* = 0.05), which is in line to post hoc analyses of clinical trial data results [6]. Interestingly, and possibly even more relevant, SBS-QoL^TM^ sum score and subscale 1 changes over time in teduglutide-treated patients changed numerically and met the previously described minimal clinically important difference (MCID) of −18.4 QoL change [6], while this difference was not shown in matched patients without treatment. A considerably longer observation period in the present study compared to the pivotal phase III trial (4.3 years vs. 24 weeks [6]) may be the major explanation for greater QoL changes, leading to statistical significance, as it was seen, that teduglutide response (and therewith a potential effect on QoL) occurs over a long time interval [16,17,18,19].

For the present study, matching criteria were well-selected according to the results of previously published research findings. Besides the fact that observation periods should generally be equal between groups, the time for follow-up is a matching criterion of major relevance in SBS-IF patients because these patients accommodate themselves, over time, to their new life situation after recovering from the initial event that caused intestinal failure [6]. Further, general PN duration [9], PS burden (represented by PN days per week [9,20], PS volume per week [10]) and bowel anatomy [5,21] were described to have an influence on quality of life and qualified to differ as minimally as possible between groups. After baseline data acquisition, bowel anatomy changed from type 2 to type 3 in a teduglutide-treated patient and from type 1 to type 2 in two controls due to reconstructive surgery. This lead to a relatively balanced anatomy composition in both groups at follow-up (12% stoma patients in the teduglutide group vs. 24% in the control group), but may have had an influence (possible improvement) on QoL, especially in the non-treated control group, which could not be corrected for and may be one explanation for why statistical significance of between-group comparison was not reached for the SBS-QoL^TM^ sum score and subscale changes.

After 4.3 years of follow-up, PS volumes were reduced significantly by −7.6 l and −4.3 l in teduglutide-treated patients and controls, respectively. Additionally, PN days per week were reduced significantly in both groups (−3.3 vs. −1.7). The workgroup of Amiot et al. reported that the actuarial PS dependence probabilities of non-teduglutide-treated cIF patients were 74%, 64% and 48% at 1, 2 and 5 years after surgery, respectively [22]. In the present study, at study inclusion, over 50% of the non-treated patients were below five and even below the first two years of a post-SBS causing event (median cIF duration of 1.5 years vs. 2.4 years in teduglutide group). Given a similar median follow-up period of 4.3 years in both groups, the timeframe from cIF onset until follow-up has a little shift to a higher weaning probability in the non-treated group vs. the teduglutide-treated group (weaning rather occurs in a timely manner after the cIF-causing event), or in other words: before study inclusion, the control group has had a little less time (−1 year) after the initial cIF-causing event to recover and to adapt spontaneously compared to the teduglutide group. This imbalance, plus the above-described changes in bowel anatomy in two individuals, may explain, at least partly, the observed PS reductions in the control group.

It was shown that PS reduction during teduglutide treatment is beneficial for QoL in SBS-cIF patients [6]. Unfortunately, ANCOVA could not be performed in the present study in order to statistically analyze the influence of PS volume reduction on QoL change, because statistical assumptions were not met. Correlation analyses for changes in QoL and changes in PS volume did not show a linear relationship, which is in contrast to previous findings [6,10]. Additionally, as PS volume reductions were seen in both groups, it might be plausible to assume that the increase in QoL is the result of the GLP-2 effect, in general, and not only the effect of PS reduction and other factors beyond PS reduction may be relevant for QoL in SBS-cIF, as described elsewhere [3]. In the present study, it was shown that stool frequency decreased over time in teduglutide-treated patients, and increased in non-treated patients, with significant differences between the groups. Additionally, stool consistency improved in teduglutide-treated patients and worsened in non-treated patients. Both variables had been analyzed before in real-world settings [16,23,24], but a direct comparison to matched non-treated patients has not been reported so far. Another potential QoL-influencing factor is sleep quality [17,24]. The number of sleep disturbances could not be analyzed longitudinally due to missing baseline data here, but it could be shown that the number of sleep disturbances per night was significantly lower at follow-up in treated vs. non-treated patients, which supports the previously found sleep-improving effect of teduglutide [17]. Having in mind that the present study compared teduglutide-treated individuals with individually matched control patients and that SBS patients suffer from high stool frequencies of up to 15 times per day [16], which may seriously impair patient’s daily life [21], especially the presented results from stool analyses further support the idea that stool improvement may contribute meaningfully to QoL improvement. 

The major limitation of the present study comes with the nature of the study, as the number of participants was limited due to the matched pair study design, and no power calculation was performed based on QoL outcomes, giving it a rather exploratory character. Plus, from study inclusion (=time of questionnaire completion) until teduglutide exposure, a median time of 2 months passed (min = 0, max = 20 months), in which QoL may also have had improved therapy, independently. Another limitation of the present study is that the SBS-QoL^TM^ questionnaire is a rather recently introduced patient-reported outcome measure and does not include a weighing of the single items according to their clinical significance for patients, yet [10]. Additionally, although several changes in QoL scores were shown to be statistically significant, and even reached the MCID, the real clinical meaningfulness remains to be evaluated by an empirical anchor-based approach, as recommended by US Food and Drug Authority [25]. Furthermore, due to the predefined dataset from the PNLiver trial, baseline data for the number of sleep disturbances are not available for longitudinal analyses; therefore, it remains unknown whether the reported inter-group difference at follow-up existed already at baseline/changed over time. Lastly, a spontaneous adaptation to an unknown extent could not only have affected the non-treated cohort, but also QoL changes in the teduglutide-treated group.

## 5. Conclusions

In conclusion, this is the first clinical real-world study that supports the assumption that teduglutide may be beneficial to the QoL of SBS-cIF patients. The real clinical meaningfulness of the extent to which QoL scores were improved remains to be evaluated within future studies.

## Figures and Tables

**Figure 1 nutrients-15-01949-f001:**
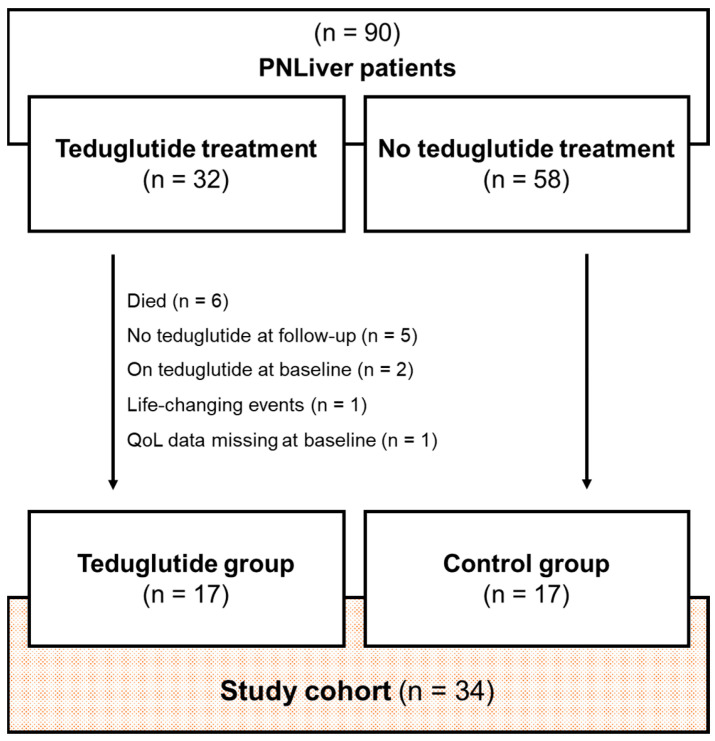
Study flow chart.

**Figure 2 nutrients-15-01949-f002:**
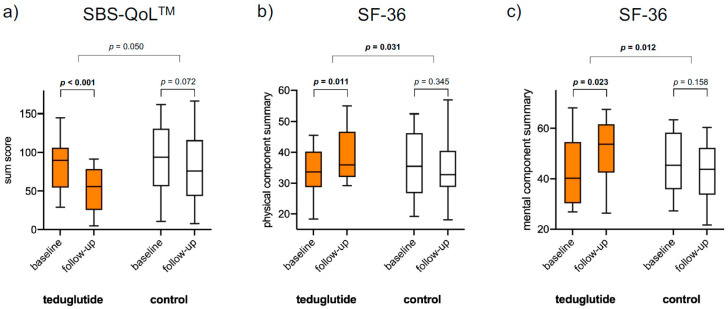
QoL changes over time and between groups. SF-36: positive changes representing QoL improvement; SBS-QoL^TM^: negative changes representing QoL improvement. Presented are means.

**Figure 3 nutrients-15-01949-f003:**
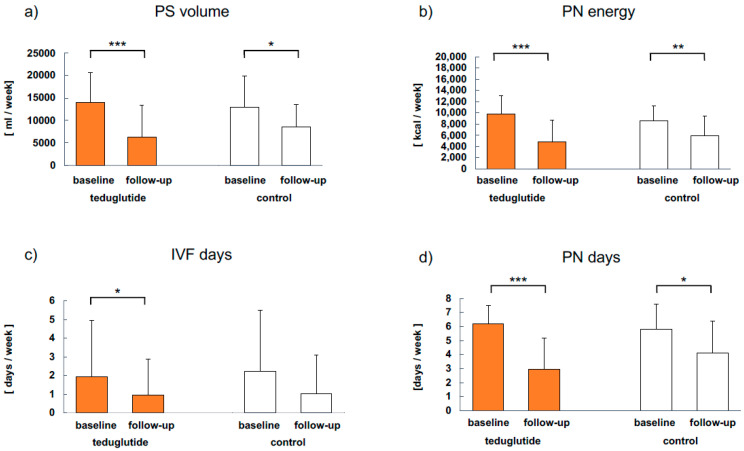
PS characteristics (*n* = 17) for teduglutide-treated patients and non-treated controls. (**a**) Parenteral support volume, (**b**) parenteral nutrition energy, (**c**) intravenous fluid days per week, (**d**) parenteral nutrition days per week. Significant changes are indicated as: * *p* < 0.05, ** *p* < 0.01 and *** *p* < 0.001 vs. baseline values. IVF, intravenous fluids; PN, parenteral nutrition; PS, parenteral support. Presented are means.

**Figure 4 nutrients-15-01949-f004:**
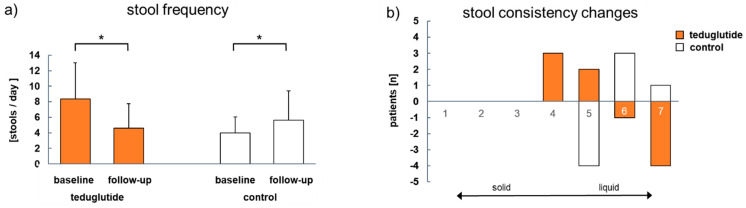
(**a**) Stool frequency per day and (**b**) stool consistency changes from baseline to follow-up according to Bristol Stool Scale (1 = separate hard lumps, 2 = sausage-shaped but lumpy, 3 = like a sausage but with cracks, 4 = like a sausage, smooth and soft, 5 = soft blobs with clear-cut edges, 6 = a mushy stool, 7 = watery), teduglutide *n* = 17; control *n* = 15. Significant changes are indicated as: * *p* < 0.05 vs. baseline values. Presented are (**a**) means and (**b**) number of patients.

**Figure 5 nutrients-15-01949-f005:**
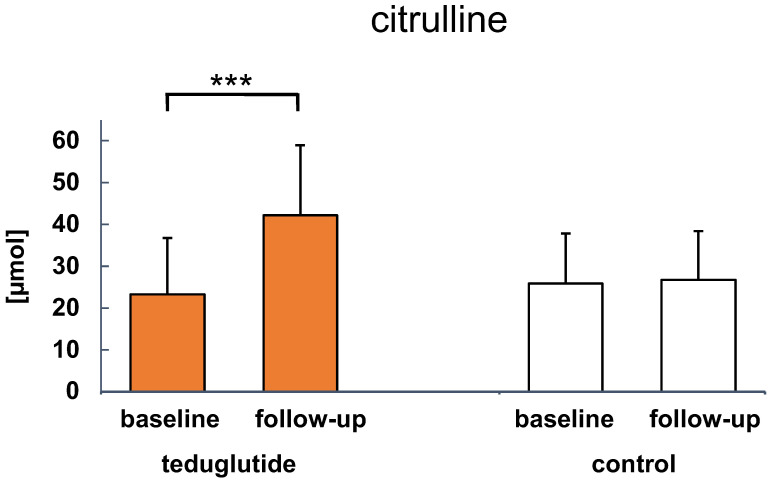
Plasma citrulline levels for teduglutide-treated patients and non-treated controls (*n* = 13 and *n* = 11; *n* = 11 and *n* = 14, respectively). Significant changes are indicated as: *** *p* < 0.001 vs. baseline values. Presented are means.

**Table 1 nutrients-15-01949-t001:** Patient characteristics for teduglutide and control group.

Patient Characteristics	Teduglutide Group	Control Group
Number of patients	17	17
Female sex, *n* (%)	9 (53%)	15 (88%)
Age (years)		
Mean ± SD	48.2 ± 20.0	49.6 ± 17.2
Median (IQR)	53.3 (36)	56.2 (28)
BMI (kg/m^2^)		
Mean ± SD	21.4 ± 3.4	20.4 ± 4.4
Median (IQR)	20.4 (5.1)	20.2 (6.8)
Cause of major intestinal resection, n (%*)		
Mesenteric ischemia	5 (29%)	7 (41%)
Inflammatory bowel disease (IBD)	3 (18%)	4 (24%)
Traumatic injury	3 (18%)	0 (0 %)
Adhesion ileus	3 (18 %)	3 (18%)
Surgical complications	1 (6 %)	2 (12%)
Other (motility disorder due to aganglionosis, benign tumor)	2 (12%)	1 (6%)
Length of remaining small bowel (cm)		
Mean ± SD	77 ± 37	111 ± 40
Median (IQR)	70 (60)	104 (36)
Presence of ileocaecal valve, n (%)	1 (6%)	2 (12%)
Colon in continuity, n (%)	15 (88%)	11 (65%)
Duration of cIF (years)		
Mean ± SD	4.9 ± 5.1	3.7 ± 4.9
Median (IQR)	2.4 (7.0)	1.5 (3.0)
Prescribed total parenteral volume (L/week)		
Mean ± SD	13.9 ± 6.6	12.9 ± 6.9
Median (IQR)	14.0 (12.7)	14.0 (9.4)
Prescribed parenteral energy (kcal/kg/d)		
Mean ± SD	26.7 ± 6.6	28.2 ± 6.7
Median (IQR)	26.8 (9.6)	27.3 (9.2)
Prescribed parenteral energy (kcal/week)		
Mean ± SD	9846 ± 3218	8615 ± 2693
Median (IQR)	10,514 (5645)	9450 (5107)
PN-days (days/week)		
Mean ± SD	6 ± 1	6 ± 2
Median (IQR)	7 (2)	7 (3)
IVF-infusions (number of fluid and electrolyte infusions/week)		
Mean ± SD	2 ± 3	2 ± 3
Median (IQR)	0 (5)	0 (7)

* Rounding error due to inexactness in the representation of real numbers. SD, standard deviation; IQR, interquartile range; PN, macronutrient containing admixture.

**Table 2 nutrients-15-01949-t002:** Matching characteristics (ranked order).

		Teduglutide Group *n* = 17		Control Group *n* = 17	
		Median	IQR	Median	IQR
	PN duration before baseline visit (years)	1.3	4.3	1.0	1.6
SBS-Qol ^TM^	Sum score	89.9	51.9	93.9	74.8
	Subscale 1	60.1	34.0	59.7	57.4
	Subscale 2	26.4	15.8	23.8	24.4
SF-36	Physical component summary	33.7	11.6	35.5	19.5
	Mental component summary	40.2	22.6	45.3	22.5
	Time to follow-up * (years)	4.3	4.6	4.3	4.8
PS burden	PN days per week	7	2	7	3
	PS volume per week (L)	14.0	12.7	14.0	9.4
Bowel anatomy	Type 1, (*n* (%**))	2 (12%)	-	6 (35%)	-
	Type 2, (*n* (%**))	12 (71%)	-	7 (41%)	-
	Type 3, (*n* (%**))	3 (18%)	-	4 (24%)	-

* Follow-up time of teduglutide group is equivalent to teduglutide treatment duration. ** Rounding error due to inexactness in the representation of real numbers. IQR, interquartile range.

**Table 3 nutrients-15-01949-t003:** QoL analyses over time and between groups. Presented are means for better comparability to previous literature. Change: follow-up to baseline. SF-36: positive changes representing QoL improvement; SBS-QoL^TM^: negative changes representing QoL improvement.

		Teduglutide Group *n* = 17				Control Group *n* = 17				*p*-Value between Groups
		Baseline	Follow-Up	Change	*p*-Value	Baseline	Follow-Up	Change	*p*-Value	
SBS-QoL^TM^	Sum score	84.1	51.0	−33.1	**<0.001**	89.7	79.2	−10.5	0.072	0.050
	Subscale 1	58.8	34.6	−24.2	**<0.001**	63.8	54.2	−9.6	0.109	0.063
	Subscale 2	25.3	16.4	−8.9	**0.002**	25.9	25.0	−0.9	0.431	0.076
SF-36	Physical component summary	33.7	39.0	5.3	**0.011**	36.1	35.1	−1.0	0.345	**0.031**
	Mental component summary	43.5	51.2	7.7	**0.023**	45.6	42.7	−2.9	0.158	**0.012**

## Data Availability

The data presented in this study are available on request from the corresponding author. The data are not publicly available due to institutional guidelines.

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
