# Peer review of "Quality of Life in Teduglutide-Treated Patients with Short Bowel Syndrome Intestinal Failure—A Nested Matched Pair Real-World Study"

_nutrients, 2023, doi:10.3390/nu15081949_

Round 1
Reviewer 1 Report
The manuscript is an important contribution in the field and adds to the body of knowledge on quality of life and chronic intestinal failure. I really like the inclusion of both the SF-36 and SBS-QOL in the study design.
In table 1 (or in the manuscript), you may wish to also report "prescribed parenteral energy" as mean daily energy intake (kcal/kg) since participants received PN 6 +/- 1 days/week. While it makes sense to look at "prescribed PS volume" as Liters/week, most clinicians address energy/protein as kcal/g per kg/day (clinically more meaningful than energy/week).
3.3 Quality of Life and Figure 2. I would re-iterate that the SF-36 scores and SBS-QOL scores are opposite (higher means better QOL in SF-36 and lower means better QOL for SBS-QOL). Hard to interpret the box and whisper figures without reminding the reader of this.
The results section (lines 219 - 223) provides data for sleep disturbances. There is no mention of sleep in the methodology. Please add. How was sleep disturbance evaluated? subjectively or did you administer a sleep questionnaire?
There are some typos and grammatical errors that will be identified by copy-editor, but here are a few:
Abstract (line 19) "trail" should be "trial"
Abstract (line 24) showed no significant changes in none of the mentioned scores: "none" should be "any"
Introduction (line 42): these chronically ill patient population: Rewrite as "this chronically ill patient population" or "these chronically ill patient populations".
Lasta paragraph of Introduction (line 62): "trail" should be "trial"
Use of PN and PS. Please define PS the first time it is used in the manuscript and clearly distinguish between them throughout manuscript.
Reviewer 2 Report
Grammatical or Typographical Errors:
- Page 11, para. 2 line 256: …’these patients adapt over time’…; could perhaps be changed to : these patients accommodate better to their…, as the term “adapt” and “adaptation” are used to describe changes in the intestinal function that decrease or resolve the need for Parenteral support.
General and specific comments:
- Very nice and well described study, with a good number of patients
- It would be nice to address the general reasons or characteristics that were considered in the selection of patients who would receive therapy with teduglutide. Why and how were they chosen.
- It would also be important or ideal to expand the description of matching criteria of the study population in regards to:
§ Diagnosis of cause of Intestinal Failure
§ Length or remaining small bowel
§ Presence of Ileo-Cecal valve
§ Continuity with Colon
- In terms of the Parenteral support, it would be important to have or mention de percentage of IV calories the patients were receiving.
- It would also be very important to mention the number of patients (if any) that in the non-treated group had shown Intestinal Adaptation and were off TPN.
- 5 patients had received teduglutide and the therapy was discontinued; why?
- Important to perhaps also disclose the dose of Teduglutide used.
- Did the patients in either group received any specific therapy to control diarrhea like Loperamide or similar and / or anti-secretory therapy in the TPN like H2-blockers
- It is a problem that sleep disturbance information was not collected at baseline, as this may represent a major factor in QoL and this should be emphasized.
- It appears that no patient in the treated group showed intestinal adaptation and came off TPN, is this correct?
- Was there any information on the possibility to control the diets in both groups?
